# Inhibition of Notch Signaling Enhances Antitumor Activity of Histone Deacetylase Inhibitor LAQ824

**DOI:** 10.3390/ijms241713660

**Published:** 2023-09-04

**Authors:** Qinglang Mei, Xiaohan Xu, Danling Gao, Yuting Xu, Jinbo Yang

**Affiliations:** 1Key Laboratory of Marine Drugs, Ministry of Education, School of Medicine and Pharmacy, Ocean University of China, Qingdao 266003, China; meiqinglang@stu.ouc.edu.cn (Q.M.); xxh3093@stu.ouc.edu.cn (X.X.); wugulan123@163.com (D.G.); yuting_xu7003@163.com (Y.X.); 2Innovation Platform of Marine Drug Screening & Evaluation, Qingdao Marine Science and Technology Center, Qingdao 266100, China

**Keywords:** histone deacetylase inhibitor, LAQ824, Notch signaling pathway, PF03084014, combination treatment

## Abstract

As a novel histone deacetylase inhibitor (HDACi), LAQ824 (LAQ) effectively inhibits the proliferation of hematological malignancies and solid tumors. However, phase II trials of LAQ in solid tumors were terminated due to dose-dependent toxicity. Furthermore, LAQ has been shown to induce the activation of the Notch signaling pathway in hematopoietic stem cells, which is associated with tumor progression and drug resistance in colon and breast cancers. Therefore, in this study, we investigated the strategy of LAQ combined with a Notch signaling pathway inhibitor to treat solid tumors. We used RT-PCR and Western blot methods to demonstrate that LAQ upregulated the Notch signaling pathway in solid tumor cell lines at the molecular level. The combination of LAQ and a Notch signaling pathway inhibitor was shown by a Chou–Talalay assay to have a synergistic effect in inhibiting solid tumor cell line proliferation in vitro. We also demonstrated that the combination of LAQ and a Notch signaling pathway inhibitor significantly inhibited the growth of tumor cells in vivo using an allograft tumor model. This study indicates that inhibition of the Notch signaling pathway provides a valuable strategy for enhancing solid tumor sensitivity to LAQ.

## 1. Introduction

The rationale for the use of HDACis as anti-cancer drugs is based on their ability to induce the hyperacetylation of histone and non-histone molecules, leading to increased differentiation, apoptosis, and cell cycle arrest in cancer cells [1]. HDACis have been used in the treatment of hematological malignancies as they exhibit good differential effects on normal and cancer cells at therapeutic doses. Several HDACis, such as romidepsin [2], belistat [3], and pabister [4], have shown good clinical results in peripheral T-cell lymphoma, cutaneous T-cell lymphoma, and multiple myeloma, respectively. In contrast, clinical trials with HDACis as single agents for the treatment of solid tumors have shown disappointing results [5]. LAQ, as a novel HDACi, displayed great potential for inhibitory activity in hematomas and solid tumors [6]. However, phase I trials of LAQ for solid tumors were terminated due to its dose-dependent toxicity, though the exact mechanisms involved are largely unknown [7]. Recent evidence has suggested that other antitumor compounds could enhance the chemosensitivity of tumor cells to LAQ by interfering with cell survival and proliferation and inducing apoptosis [8]. Some studies have identified the LAQ-induced upregulation of the Notch signaling pathway in hematopoietic stem cells through transcriptome data analysis, but the specific mechanism by which LAQ activates the Notch signaling pathway, as well as the Notch signaling pathway profile of LAQ processing in tumor cells, have not been investigated in depth [9,10].

The Notch signaling pathway is widespread in vertebrates and non-vertebrates as it was highly conserved throughout evolution [11]. It regulates the differentiation and development of cells, tissues, and organs by interacting with cells expressing single-channel transmembrane Notch receptors and adjacent cells expressing transmembrane Notch ligands [12,13]. During Notch signaling pathway maturation, proteolytic cleavage is triggered by furan-cleaved heterodimers in the Golgi apparatus, and the Notch receptor is transported to the cell surface [14,15]. After the Notch receptor binds to the ligand, it is cleaved into two fragments under the action of metalloproteinase (ML)/tumor necrosis factor-α converting enzyme (TACE) [16]. The lysate is further cleaved in the transmembrane region by a high-molecular-weight and multi-protein complex, which is mainly composed of γ-secretase [17]. Then, the Notch intracellular domain (NICD) enters the nucleus, binds to the CBF1/RBP-Jκ in mammals (CSL), and recruits nuclear transcription activator protein family mastermind-like (MAML) to form a ternary complex transcription activator (NICD-CSL-MAML) [18,19], which promotes downstream gene expression, thereby promoting cell proliferation and inhibiting cell differentiation.

Overexpression of the Notch signaling pathway has been found in solid cancers, such as breast cancer [20], gastric cancer [21], pancreatic cancer [22], and colon cancer [23]. Experimental evidence has indicated that targeting the Notch signaling pathway might become a new treatment method for treating cancers and overcoming tumor cells’ acquired resistance to therapy [24,25]. PF03084014 (PF), a γ- secretase inhibitor, could suppress the reduction in NICD and improve the antitumor effects of solid tumor therapies [26]. Traditional chemotherapeutic drugs, such as oxaliplatin, adriamycin, 5-fluorouracil, and gemcitabine, result in the high expression of the Notch pathway, which affects the response of tumor cells to these drugs [27]. In particular, oxaliplatin activates the Notch pathway in colon cancer cells by inducing NICD and blocking Notch signaling to increase the sensitivity of tumor cells to oxaliplatin [28]. Downregulation of the Notch signaling pathway through multiple channels is a novel strategy to increase the sensitivity of cancer cells to conventional chemotherapeutic agents.

LAQ demonstrates dose-dependent toxicity in the treatment of solid tumors [29]. The Notch signaling pathway has been found to be upregulated with LAQ treatment in hematopoietic stem cells [10]. In this report, we first identified the upregulation of the Notch signaling pathway after LAQ treatment in tumor cell lines. We also investigated the role of the Notch signaling pathway in shaping the efficacy of LAQ against solid tumors. The mechanism of action at hand was explored by assessing the combined effect of LAQ and a Notch signaling pathway inhibitor on solid tumor cell lines. The Notch signaling pathway can be used as an approach for enhancing LAQ sensitivity. Our findings provide a new perspective for the application of LAQ to the treatment of solid cancers and present potential combinational treatment strategies to enhance the treatment of solid tumors with HDACi drugs.

## 2. Results

### 2.1. HDACi LAQ Inhibits Growth of Cancer Cell Lines

LAQ has been reported to be the HDACi of the isohydroxamic acid structure that is the reciprocal isomer of hydroxamic acid (Figure 1A). First, we screened for the sensitivity of solid tumor cells to LAQ824 using the genomics of drug sensitivity in the cancer (GDSC; www.cancerRxgene.org version 8.4) database. LAQ824 showed low half-maximal inhibitory concentration (IC_50_) values in most solid tumor cells, especially in breast cancer, lung cancer and colorectal malignancies (Figure 1B). Cell viability was assessed by applying linear dosages of LAQ to different solid cancer cell lines, including breast cancer, lung cancer, and colorectal malignancies, to verify the killing effect on solid tumor cells. Compared to the HCT116 cell line, the other cell lines displayed insensitivity to LAQ, and the IC_50_ of LAQ was more evenly distributed (Figure 1C and Appendix A). Therefore, we chose the HCT116 cell line (IC_50_ = 0.05 μM) for subsequent experiments. Furthermore, colony-formation assay analysis of the HCT116 cell line indicated that LAQ could inhibit long-term tumor cell proliferation (Figure 1D). We observed that the effect of killing tumor cell lines of LAQ were obvious in vivo and in vitro with toxicity to the organism (Appendix A). LAQ was reported to be an effective HDACi [30]. The effectiveness of histone deacetylase suppression was determined by evaluating the expression of histone acetylation and *P21* in the HCT116 cell line. LAQ (0.05 μM) was applied to the HCT116 cell line for different periods of time. The results showed *P21* activation and increased acetylation of histone 3 (Ac-H3) occurred immediately after 1 h of treatment (Figure 1E,F). A dose-dependent increase in histone hyperacetylation and *P21* caused by LAQ was observed (Figure 1G,H). These results indicated that LAQ upregulated histone acetylation and exerted a killing effect on the colon cancer cell line.

### 2.2. Notch Signaling Pathway Activated after LAQ Treatment in HCT116 Cell Line

Based on the available findings, LAQ activated the Notch signaling pathway in hematopoietic stem cells. The overexpression of Notch receptors and their ligands has been proven to play a crucial role in tumor formation, as well as metastasis in multiple cancer types, such as lung, breast, and colon cancer [31]. In some malignancies, activation of the Notch signaling pathway is a key mechanism of acquired resistance to chemotherapeutic medicines [32]. Thus, we explored the stimulation of the Notch signaling pathway after LAQ treatment in the colorectal tumor cell line. This study determined the effect of LAQ on intracellular levels of the Notch signaling pathway in the HCT116 cell line. Hes family bHLH transcription factor 1 (Hes1), a downstream target gene of the Notch signaling pathway, was assessed during 24 h exposure to LAQ and PF. Results showed that Hes1 was significantly increased in the LAQ group compared with the untreated group (Figure 2A). Furthermore, activation of the Notch signaling pathway was detected by the mRNA expression of Hes1 and Hes-related family bHLH transcription factor with YRPW motif 1 (*Hey1*) treated with LAQ in the HCT116 cell line. Our research revealed that when the concentration of LAQ reached 10 nM and the LAQ treatment lasted for one hour, the expression of *Hey1* and *Hes1* appeared to rise (Figure 2B,C). The activation of Notch signaling largely depends upon NICD, a key protein in the classical Notch signaling pathway. Then, we examined the possible mechanism of increased NICD by treating cells with LAQ for the indicated periods (0, 1, 2, 4, 6, 8 h) and with different concentrations (0, 0.01, 0.05, 0.1, 0.2 μM). The expression of NICD was induced in <1 h and at 10 nM after LAQ stimulation (Figure 2D,E). Therefore, we hypothesized that accumulated NICD was not caused by increased transcription but might be caused by increased cleavage by ADAM family proteases or γ- secretase. We determined that PF03084014 (PF), a γ- secretase inhibitor, could suppress the upregulation of NICD by LAQ and improve the antitumor effects [26]. The expression of *Hes1* and *Hey1* in the HCT116 cell line, which was incubated with LAQ and PF for 24 h, was decreased compared with that in cells incubated with LAQ alone (Figure 2F). These results indicated that LAQ activated the Notch signaling pathway in the colorectal cancer cell line.

### 2.3. Combined Treatment of LAQ and PF Synergistically Reduced Cell Proliferation in the HCT116 Cell Line

We investigated whether dual inhibition showed any additive effects on cell proliferation. Cell lines were treated with LAQ alone or combined with PF for 48 h and cell viability was measured using an MTT assay. We compared the effects of a combination treatment with an individual treatment of LAQ or PF and calculated the combination index (CI) using Compusy software [33]. The results showed that the combined inhibition of proliferation was synergistic at a CI < 1 in the HCT116 cell line (Figure 3B). The combination of PF and LAQ decreased viability significantly compared with either treatment alone (Figure 3A). These results suggest that PF could sensitize the cancer cell line to LAQ. In addition, to confirm the inhibitory effect of the combination of PF and LAQ on the long-term growth potential in tumor cells, a colony formation assay was performed in the HCT116 cell line. As the results display, LAQ or PF alone had a limited inhibitory effect on tumor cell growth. However, cell growth was dramatically suppressed after combined treatment in the HCT116 cell line (Figure 3C).

### 2.4. Combined Treatment of LAQ and PF Synergistically Inhibited Invasion in HCT116 Cell Line

Studies have revealed that the aggressiveness of tumor cells indicates a poor prognosis [34]. To confirm the invasive modification in the HCT116 cell line following exposure to LAQ and PF, an evaluation was conducted with a scratch assay. The scratch wounds and the migration cells in the scratch were photographed at 48 h after scratching, and the average percentage of scratch recovery was determined. The results displayed that the PF and LAQ combination could significantly inhibit HCT116 cell line migration (Figure 4A). Matrix metalloproteinase-2 (MMP2) and matrix metalloproteinase-9 (MMP9) are matrix metalloproteases involved in cancer progression and invasion, and their expression is considered a marker of cell invasiveness [35]. We performed Western blot analysis to test the levels of MMP2 and MMP9. The results showed that although single treatment reduced invasion, the invasive ability was further reduced by dual-pathway inhibition (Figure 4B).

### 2.5. Combined Treatment of LAQ and PF Synergistically Promoted Apoptosis and G2/M Phase Arrest in HCT116 Cell Line

Research has revealed that LAQ can inhibit cell growth and apoptosis in tumor cells [36]. To determine whether inhibition of the Notch signaling pathway enhances cell cycle arrest, we stained the HCT116 cell line with propidium iodide (PI) after LAQ and PF treatment for 24 h and then analyzed these cells using flow cytometry. LAQ induced cell G2/M phase arrest in the HCT116 cell line while PF induced cell G1 phase arrest (Figure 5B). Furthermore, it was confirmed that the combination of LAQ and PF enhanced cytotoxicity via apoptosis induction using annexin V-FITC/PI flow cytometry analysis. Compared to apoptosis induced by either LAQ or PF alone, the combination of LAQ and PF led to significantly higher apoptosis in the cancer cell line (Figure 5A).

### 2.6. Combined Treatment Reduced Tumor Growth and Synthetic Lethality In Vivo

To further investigate the potential efficacy of combined LAQ and PF inhibition, we performed a preclinical therapeutic drug trial in vivo using a xenograft tumor model of the HCT116 cell line in nude mice. According to the research previously published and the metabolic dynamics of the chemicals, we selected an intravenous dose scheme of 10 mg/kg of LAQ and an oral dosage of 100 mg/kg of PF, with their respective tumor growth inhibition rates up to 40% [29]. LAQ and PF treatment was performed in the general scheme (Figure 6A). Body weight (Figure 6C) and tumor volume were measured every two days during the administration period (Figure 6D). At the end of treatment, all the mice were killed. Freshly removed tissue samples were split and photographed (Figure 6B). The antitumor activity of LAQ and PF in a xenograft tumor model was assessed with tumor volume analysis. The results showed that LAQ (10 mg/kg) delayed tumor growth, and the inhibition rate for tumor development was 43.90%. When used in combination, the LAQ (10 mg/kg) and PF (100 mg/kg) inhibitiory effect on tumor growth was 63.69% (Figure 6E), equal to the inhibitory effect of LAQ (15 mg/kg) (Appendix A). Western blot analysis was performed to detect the efficacy of the inhibitors. In the LAQ group, both NICD and Ac-H3 were overexpressed. In addition, PF treatment alone decreased NICD translocation (Figure 6F), while combination treatment decreased NICD and Hes1 compared with the other groups (Figure 6F). Furthermore, the apoptotic protein expression was detected using Western blot assay. LAQ or PF individually had limited effects on the expression of B cell lymphoma-2 (BCL2), an anti-apoptotic protein. However, a dramatic suppression of BCL2 was observed after the combined treatment of PF and LAQ, which inhibits anti-apoptotic protein expression to promote apoptosis in tumor cells (Figure 6G). Ki67 indicates malignant tumor proliferation, and its expression correlates with tumor prognosis [37]. IHC results showed that Ki67 was reduced in tumors from HCT116 xenografts treated with LAQ and further downregulated by combination treatment with PF (Figure 6H). These results indicate the potential value of LAQ and its combined use with a Notch inhibitor in colorectal cancer.

## 3. Discussion

HDACi exerts direct antitumor effects by regulating the acetylation status in tumor cells, inducing apoptosis, blocking the cell cycle, promoting DNA damage, and inducing autophagy. HDACi also affects other signaling pathways through epigenetic pathways, thus influencing tumor cells [1]. To address the resistance and toxic side effects of currently marketed HDACis, researchers have achieved better solutions by combining them with other antitumor compounds [38]. As such, exploring appropriate combination regimens can significantly improve the antitumor activity of HDACis [39]. LAQ has significant repressed solid and hematological tumor effects as a new type of HDACi, giving it a significant inhibitory effect on solid and hematologic tumors. LAQ can activate the *P21* promoter in the cell to increase its expression, which is also consistent with our experimental results [6]. *P21* is one of the downstream targets of Notch, and whether its increase is related to Notch requires further exploration.

The Notch signaling pathway is activated in many cancers and affects tumor cell growth, development, and metastasis. At the same time, Notch receptors and ligands have been discovered to be prognostic markers of human cancer. In recent years, increasing studies have found that the Notch signaling pathway may play an important role in regulating anti-cancer drug sensitivity and acquired resistance [40]. Targeting the Notch signaling pathway is a new combination therapy method that might overcome the resistance of cancer cells.

Previously, activation of the Notch signaling pathway was correlated with the process of cancer and the possibility of drug resistance. So, a study treating tumor cells with LAQ was conducted to investigate the activation of the Notch signaling pathway. Our study utilized Western blot and RT-PCR assays to test the expression of *Hes1* and *Hey1* treatment with LAQ in colorectal and breast cancer cell lines. The results revealed that the expression of *Hes1* and *Hey1* was upregulated in colorectal and breast cancer cell lines. Additional research is required to investigate the transduction of the Notch signaling pathway when treated with other HDACis. Then, the sensitivity of tumor cells to LAQ and PF, indicating the inhibition of the Notch signaling pathway, was detected. PF is a selective inhibitor of γ-secretase that has demonstrated substantial antitumor activity in the Notch signaling pathway [26]. Our results indicated that the combination of PF and LAQ could synergistically reduce the viability of colorectal and breast cancer cell lines (Appendix A). This effect was related to reducing the production of NICD and inhibiting the activation of Hes1. The combination of PF and LAQ may provide a new method with which to enhance tumor chemotherapy. There is a need for more research to explain the precise mechanism of combination therapy in cancers.

Previous studies have found that using LAQ alone made cells stagnate in the G2/M phase. PF could sensitize tumor cells to LAQ. After PF was introduced, cell cycle arrest and apoptosis both increased. Further study of the antitumor activity of LAQ in vivo revealed that combining with PF significantly enhanced the antitumor activity of LAQ. The combination of PF and LAQ can increase the antitumor effectiveness of LAQ by preventing Notch activation. As a result, it may be feasible to increase the druggability of a chemical by enhancing its efficacy and reducing the concentration of LAQ employed in combination with PF during the drug development process. 

This study has guiding significance for the research and development process of antitumor drugs, and it opens up a new direction in the combination therapy of HDACis treat to cancer. Previously, multiple studies demonstrated a correlation between drug resistance in tumors and the activation of Notch signaling. Activation of the Notch signaling pathway provides new inspiration for research into the toxic effects of HDACis. The relationship between the toxic effects of HDACis and the activation of Notch signaling can be further investigated. Numerous HDACis were found to upregulate the Notch signaling pathway, such as LBH589 (Panobinostat) [41], valproic acid (VPA), and suberoyl bishydroxamic acid (SBHA) [42], which express a stronger antitumor effect. To build on this, HDACi and Notch inhibitor combinations should be explored for more effective combined therapies. Additionally, the effectiveness of HDACis with Notch inhibitors against multidrug-resistant bacteria should be investigated further. 

## 4. Materials and Methods

### 4.1. Cell Culture and Reagents

Cell lines HCT-116, SW480, DLD-1, HCT-15, RKO, HT29, MB-MDA-231, MCF-7, and A549 were obtained from the American Type Culture Collection. Cells were cultured in growth medium (Gibco, Grand Island, NY, USA) as recommended by the respective suppliers supplemented with 10% fetal bovine serum (Gibco, Grand Island, NY, USA), 100 IU/mL penicillin and 100 mg/mL streptomycin, at 37 °C and containing 5% carbon dioxide. LAQ (Dacinostat, Catalog No.S1095) and PF03084014 (Nirogacestat, Catalog No.S8018) were obtained from Selleckchem (Houston, TX, USA).

### 4.2. Animals

All animal studies were approved and supervised by the Ocean University of China Animal Care and Use Committee, and all procedures were performed in accordance with the Ocean University of China Institutional Animal Welfare Guidelines (OUCSMP-20211201). All mice were housed in a specific pathogen-free (SPF) animal facility with a 12 h day/night cycle. Female BALB/c-nu mice (6 weeks old) were obtained from Beijing Viton Lever (Beijing, China).

### 4.3. Cell Viability

For the cell viability assay, 3000 cells were seeded in 96-well plates for 8 h. Then, the cells were treated with either a vehicle or compounds. Next, 20 μL of MTT (5 mg/mL) was added to each well after 48 h and the cells were incubated for another 4 h. The supernatant was removed and the precipitate dissolved using DMSO. The absorbance was measured at 490 nm using SpectraMax@i3.

### 4.4. Western Blotting

For Western blotting, cells were washed with cold PBS and lysed with cell lysis buffer containing 1× protease inhibitor and 1× phosphatase inhibitor (Cell Signaling Technology, CST, Danvers, MA, USA) for 30 min on ice. Then, the cell lysates were centrifuged at 12,000× *rcf* for 10 min at 4 °C. The protein concentrations were detected with a BCA assay. The protein lysates were separated using SDS-PAGE and transferred to PVDF membranes. After adding the indicated antibodies, immune complexes were detected with HRP substrate (Millipore, Burlington, MA, USA) and photographed with a Tanon 5200 imaging system. Antibodies against NICD, Hes1, and Ac-H3 were obtained from CST, and antibodies against α-tubulin were obtained from Santa Cruz (Santa Cruz, CA, USA).

### 4.5. Flow Cytometry Assays

For flow cytometry, 3 × 10^5^ HCT116 cells were plated in 6-well plates for 8 h and treated with a vehicle, PF, or LAQ at the indicated concentrations for 24 h. Then, the cells were washed with cold PBS and harvested using trypsinization. Cell staining was performed according to the protocols of the Cell Cycle Staining Kit (Lianke Bio, Hangzhou, China, Cat#: CCS012) and the Annexin V-FITC/PI apoptosis kit (Lianke Bio, Cat#: AP101-100). Cells were detected with flow cytometry (FACS Arial Ⅲ BD).

### 4.6. Immunohistochemistry

Tumor specimens were fixed in 10% neutral buffered formalin for 24 h, followed by standard tissue processing and embedding. Sections were cut at 4 μm and dried overnight at 37 °C under a microscope. Tissue sections were stained with a primary antibody for Ki67 (ab92742, Abcam, Hongkong, China) overnight, and a secondary antibody was incubated for 30 min. All sections were nuclear stained with hematoxylin, dehydrated, cleared, and examined microscopically.

### 4.7. Colony-Formation Assays

HCT116 cell lines were housed at a density of 700 cells per well in 6-well plates containing 2 mL of complete medium, and then cultured at 37 °C and 5% CO_2_. After administration of the inhibitor or DMSO for 48 h, the cells were cultured for 10 days. Cell colonies were stained with 0.1% crystalline violet and analyzed with microscopy.

### 4.8. HCT116 Xenograft Tumor Model and Administration of Therapeutic Inhibitors

Mice were injected subcutaneously with 5 × 10^6^ HCT116 cells (0.1 mL) suspended in a serum-free medium. When the tumor volume reached 40–100 mm^3^, mice were randomly assigned to 5 groups. The first group was given an injection of solvent as a blank control, and the second group was given a tail vein injection of 5-FU (20 mg/kg), administered as frequently as every three days. The third group was given LAQ (10 mg/kg) by tail vein injection, administered as frequently as once per day, five times a week. The fourth group was given PF-03084014 (100 mg/kg) administered by gavage twice per day for three consecutive days with a four-day stop. The fifth group received LAQ (10 mg/kg) combined with PF03084014 (100 mg/kg). During the administration period, body weight was recorded and the tumor volume was measured every two days, calculated using the following formula: volume = (L × W^2^)/2, where L equals length and W equals width. At the end of treatment, all mice were killed. Freshly removed tissue samples were split; some were fixed in 5% paraformaldehyde solution and others placed in protein lysate for further analysis.

### 4.9. Statistical Analysis

All the experiments were conducted in triplicate. GraphPad Prism software version 8 was used to statistically analyze the data. *p* values were calculated using Student’s *t*-test and were considered statistically significant when *p* <0.05. Combination indexes (CIs) were analyzed with the Chou–Talalay algorithm method. The CIs for all assessed combinations are shown. Synergism, additive effects, and antagonism were defined as CI < 1, CI = 1, and CI > 1, respectively.

## 5. Conclusions

In this project, we explored the molecular mechanisms of LAQ, which can effectively kill solid tumor cells. The dynamic activation of Notch signaling provides a new perspective for understanding the unsatisfactory effect of HDACis. Our research offers new insight into the potential of Notch as a combination strategy with LAQ in colorectal and breast cancers. Further research is needed to confirm these findings and translate them into clinical practice.

## Figures and Tables

**Figure 1 ijms-24-13660-f001:**
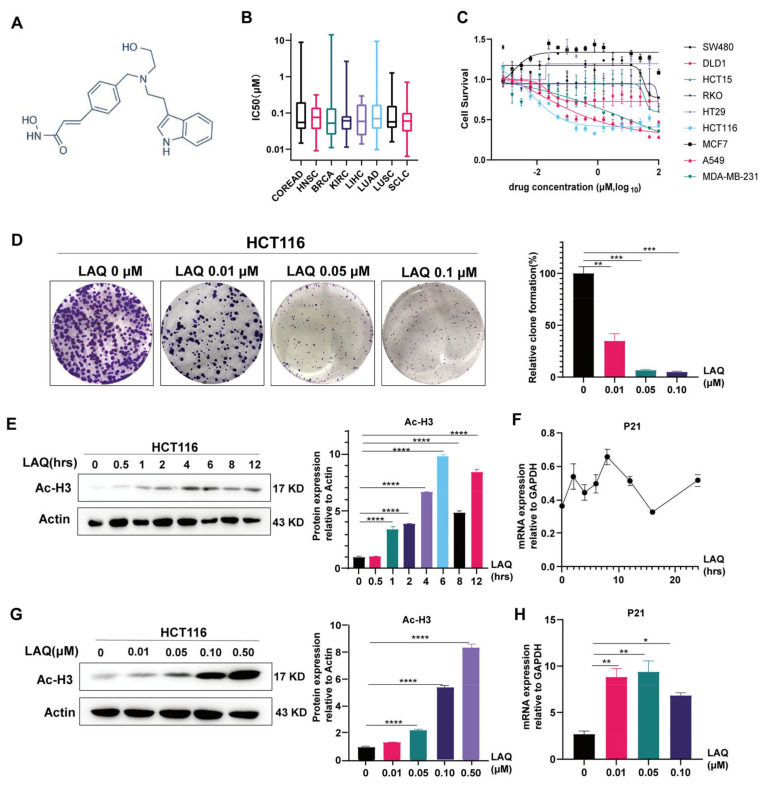
HDACi LAQ inhibits proliferation of cancer cell lines. (**A**) The chemical structure of LAQ; (**B**) IC50 values of LAQ824 in 8 solid tumors. We screened out LAQ with low IC50 values in 279 solid tumor cell lines from the genomics of drug sensitivity in the cancer database and found that LAQ inhibited the viability of most solid tumor cell lines. (**C**) MTT assay detects proliferation ability of nine different cell lines after 48 h of treatments with different concentration of LAQ (100, 50, 25, 12.5, 6.25, 3.125, 1.5625, 0.7812, 0.3906, 0.1953, 0.0976, 0.0488, 0.0244, 0.0122, 0.0061 03516, 0.0030, 0.0015, 0.0007 μM). (**D**) Colony formation assay detects long proliferation of HCT116 cell line treatment with range dose of LAQ. (**E**) Western blot analysis of the Ac-H3 in HCT116 cell line with LAQ (0.05 μM) for different times; total actin was similarly analyzed. (**F**) The expression of *P21* was detected by qRT-PCR after treatment with LAQ (0.05 μM) for different times. (**G**) Western blot analysis of the Ac-H3 in HCT116 cell line with different-concentration LAQ for 24 h; total actin was similarly analyzed. (**H**) The expression of *P21* was detected by RT-PCR after treatment with different-concentration LAQ for 8 h. The data are expressed as the mean ± SD. Significant differences were analyzed using the unpaired two-tailed *t*-test. * *p* < 0.05, ** *p* < 0.01, *** *p* < 0.001, **** *p* < 0.0001. *n* = 3.

**Figure 2 ijms-24-13660-f002:**
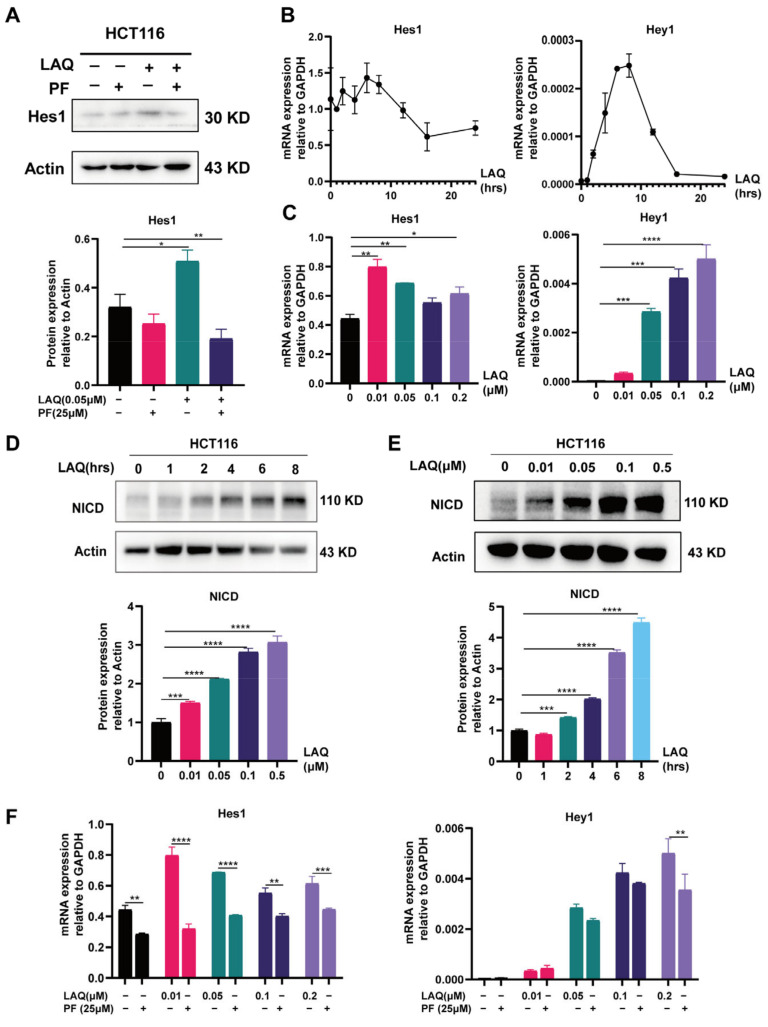
Notch signaling pathway activated after treatment with LAQ in HCT116 cell line. (**A**) The expression of Hes1 was assessed by Western blot after treatment with LAQ and PF. (**B**) The expression of *Hes1* and *Hey1* was detected by qRT-PCR after treatment with LAQ for different times in HCT116 cell line. (**C**) The expression of *Hes1* and *Hey1* was detected by qRT-PCR after treatment with different-concentration LAQ in HCT116 cell line. (**D**) Western blot analysis of the NICD in HCT116 cell line with LAQ for different concentrations; total actin was similarlt analyzed. (**E**) Western blot analysis of the NICD in HCT116 cell line with LAQ for different times; total actin was similarly analyzed. (**F**) The expression of *Hes1* and *Hey1* was detected by qRT-PCR after treatment with LAQ and PF in HCT116 cell line. The data are expressed as the mean ± SD. Significant differences were analyzed using the unpaired two-tailed *t*-test. * *p* < 0.05, ** *p* < 0.01, *** *p* < 0.001, **** *p* < 0.0001. *n* = 3.

**Figure 3 ijms-24-13660-f003:**
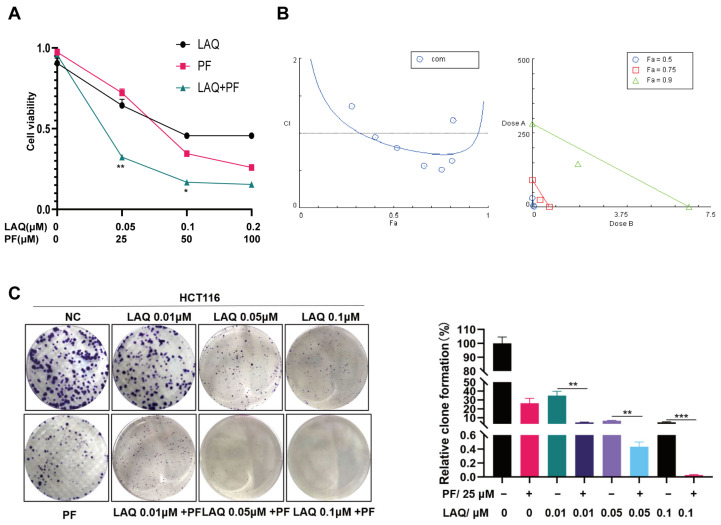
Combined treatment of LAQ and PF synergistically reduced cell proliferation in HCT116 cell line. (**A**) MTT assay detects proliferation ability after 48 h of treatments with LAQ and PF in HCT116 cell line. (**B**) CI was assessed using Compusy software after treatment with LAQ and PF for 48 h in HCT116 cell line. (**C**) Colony formation assay detects long proliferation of HCT116 cell line treatment with range dose of LAQ and PF. The data are expressed as the mean ± SD. Significant differences were analyzed using the unpaired two-tailed *t*-test. * *p* < 0.05, ** *p* < 0.01, *** *p* < 0.001. *n* = 3.

**Figure 4 ijms-24-13660-f004:**
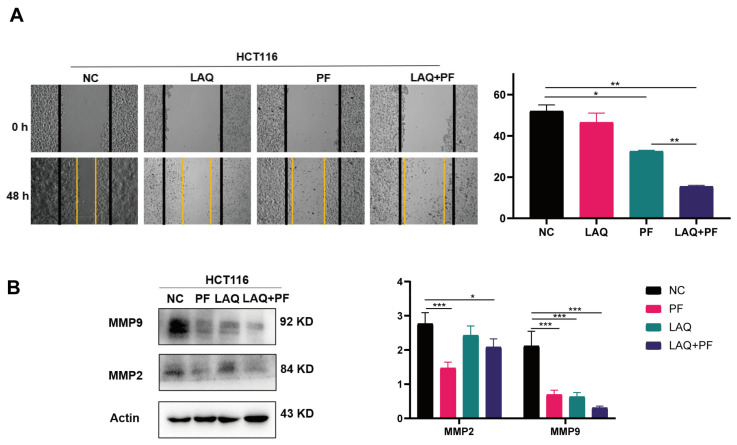
Combined treatment of LAQ and PF synergistically inhibited invasion in HCT116 cell line. (**A**) Scratch assays were conducted in HCT116 cell line with treatment of LAQ and PF. The yellow lines represent areas of cell migration. (**B**) Western blot analysis of the expression of MMP2 and MMP9 in HCT116 cell line with treatment of LAQ and PF; total actin was similarly analyzed. The data are expressed as the mean ± SD. Significant differences were analyzed using the unpaired two-tailed *t*-test. * *p* < 0.05, ** *p* < 0.01, *** *p* < 0.001. *n* = 3.

**Figure 5 ijms-24-13660-f005:**
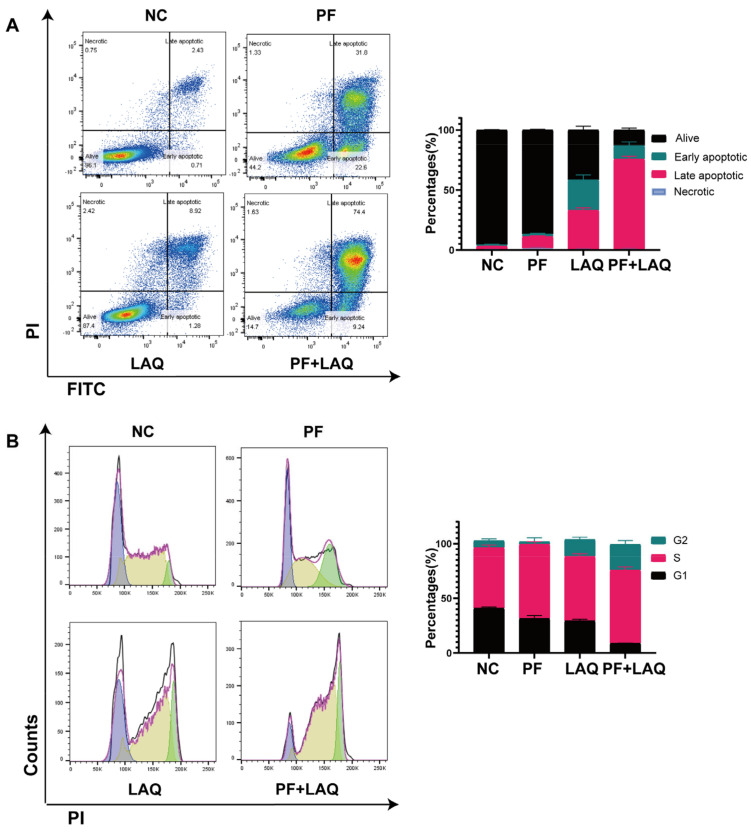
Combined treatment of LAQ and PF synergistically promoted apoptosis and G2/M phase arrest of HCT116 cell line. (**A**) Apoptosis (annexin V^+^/DAPI^−^ plus annexin V^+^/DAPI^+^) of HCT116 cell line treated with LAQ and PF was detected by flow cytometry. (**B**) HCT116 cell line was treated with LAQ and PF for 24 h, then fixed, stained with PI, and analyzed using flow cytometry. Blue represents G1, yellow represents S phase, green represents G2 phase, and purple represents total. The data are expressed as the mean ± SD. *n* = 3.

**Figure 6 ijms-24-13660-f006:**
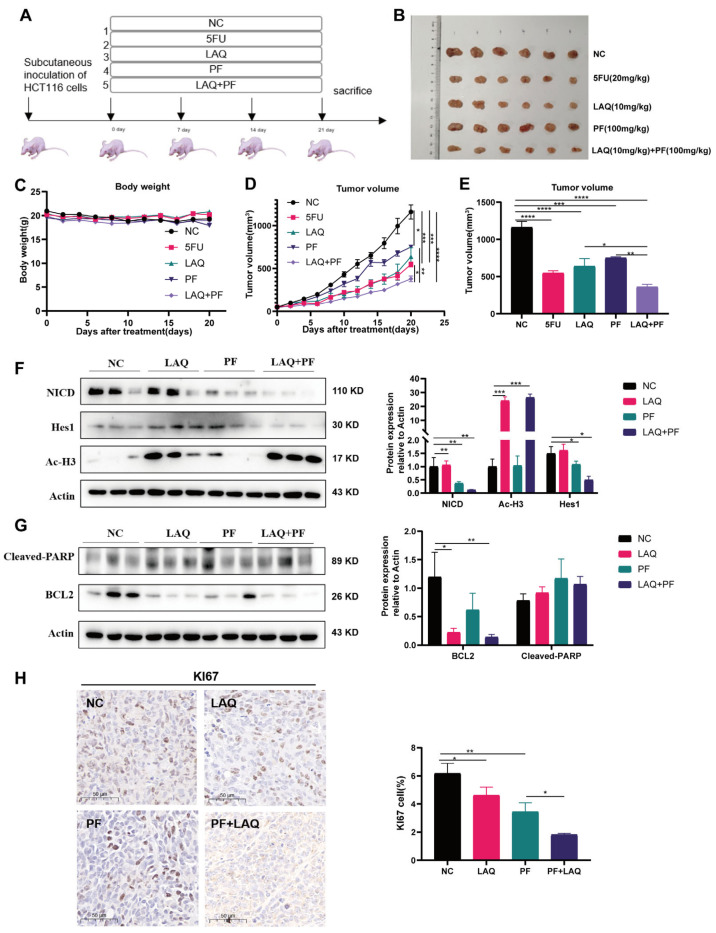
Combined treatment reduced tumor growth and synthetic lethality in vivo. (**A**) LAQ and PF treatments were performed as outline in the general scheme. (**B**) Photographic records of the tumors in each group described above that were removed on day 21. (**C**) Recording of the body weight in each group during gavage. (**D**) Expression of the tumor volume in each group during gavage. (**E**) Illustration of the tumor volume in each group described above for tumors that were removed on day 21. (**F**) Tumor tissue sections were subjected to Western blot analysis of NICD and Ac-H3 by grouping. Quantification of relevant proteins was statistically analysed. (**G**) Tumor tissue sections were subjected to Western blot analysis of Cleaved-PARP and BCL2 by grouping, which were tumor apoptosis-related proteins. Quantification of relevant proteins was statistically analysed. (**H**) Representative immunohistochemical (IHC) staining for KI67 in tumor tissue, which was an indicator of tumor malignant proliferation. Quantification of relevant proteins was statistically analysed. The data are expressed as the mean ± SEM. Significant differences were analyzed using the unpaired two-tailed *t*-test. * *p* < 0.05, ** *p* < 0.01, *** *p* < 0.001, **** *p* < 0.0001. *n* = 5.

## Data Availability

Not applicable.

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
