# Peer review of "Inhibition of Notch Signaling Enhances Antitumor Activity of Histone Deacetylase Inhibitor LAQ824"

_ijms, 2023, doi:10.3390/ijms241713660_

Round 1
Reviewer 1 Report
This research investigates the potential of HDACi LAQ824 as an anti-cancer drug specifically for solid tumors. The study demonstrates that LAQ824 effectively inhibits tumor cells and activates the Notch signaling pathway, which could open up new possibilities for combination therapy. The synergistic effects observed when combining LAQ824 with the Notch pathway inhibitor PF03084014 suggest a promising approach for more effective cancer treatment.
The findings contribute valuable insights to the field of cancer drug development, and the combination approach holds promise for overcoming drug resistance in solid tumors.
This study presents original research where first-time authors investigated the potential of a combination therapy using HDACi and NOTCHi for solid tumors. Prior to this study, there was only a brief report mentioning pretreatment of NOTCHi with HDACi in an in vitro cell line as a supplementary result 1. The authors conducted comprehensive in vitro and in vivo experiments to thoroughly study the anti-cancer properties of this combination therapy. While the use of LAQ itself raises concerns about toxicity, the study's results provide strong evidence for the potential of HDACi and NOTCHi as a novel therapeutic agent.
Although it was reported that LAQ824 was well tolerated at doses that induced accumulation of histone acetylation, LBH589 (Panobinostat) is closely related structurally to LAQ824 and is more potent with better bioavailability 2. Multiple clinical trials of LBH589 (Panobinostat) are now underway.
Like LAQ824, enhanced compound LBH589, and other HDAC inhibitors such as valproic acid (VPA) and suberoyl bis-hydroxamic acid (SBHA), it has been shown that they activate Notch-1 1,3. Therefore, the contrasting role of HDACi in inducing proliferation through NOTCH signaling can be addressed by combining NOTCHi and HDACi, adding novelty and relevance to this study.
To further enhance the study, the authors could consider using the enhanced compound LBH589 (Panobinostat) with NOTCH inhibitors. However, that would be too much for sub-figures and could be explored in future independent studies.
Based on other reviewer’s feedback, it is evident that there are concerns about the direct relevance of LAQ in clinical trials. To address this, I recommend toning down the claims regarding LAQ's direct therapeutic purpose and clinical trial application. Instead, the discussion section should primarily focus on HDACi and explore how NOTCHi inhibition can be achieved, utilizing the references provided 1,3. By emphasizing HDACi and exploring the mechanisms of NOTCH inhibition using the provided references, the study can maintain its novelty and relevance.
Therefore, I suggest a revised discussion section that centers on the use of HDACi and how NOTCH inhibition can be achieved, considering the existing drug LBH589 (Panobinostat) and NOTCH inhibitors. This approach will help address readers' concerns and highlight the potential benefits of the combination therapy in a HDACi centric manner.
Despite the concerns about LAQ and the possibility of its exclusion from clinical trials, the study's broader relevance remains impressive. However, it is crucial for the authors to provide appropriate references at line 44-46. The authors conducted rigorous and carefully designed experiments both in vitro and in vivo. The quality of data and its representation are commendable, leading to a recommendation for publication.
References:
1. He, Y., Xu, L., Feng, J., Wu, K., Zhao, Y., and Huang, H. (2020). HDAC Inhibitor LBH589 Suppresses the Proliferation but Enhances the Antileukemic Effect of Human gammadeltaT Cells. Mol Ther Oncolytics 18, 623-630. 10.1016/j.omto.2020.08.003.
2. de Bono, J.S., Kristeleit, R., Tolcher, A., Fong, P., Pacey, S., Karavasilis, V., Mita, M., Shaw, H., Workman, P., Kaye, S., et al. (2008). Phase I pharmacokinetic and pharmacodynamic study of LAQ824, a hydroxamate histone deacetylase inhibitor with a heat shock protein-90 inhibitory profile, in patients with advanced solid tumors. Clin Cancer Res 14, 6663-6673. 10.1158/1078-0432.CCR-08-0376.
3. Adler, J.T., Hottinger, D.G., Kunnimalaiyaan, M., and Chen, H. (2008). Histone deacetylase inhibitors upregulate Notch-1 and inhibit growth in pheochromocytoma cells. Surgery 144, 956-961; discussion 961-952. 10.1016/j.surg.2008.08.027.
Author Response
Dear reviewer:
Thanks very much for the comments to us and for allowing us to revise our manuscript entitled “Inhibition of Notch Signaling Enhances Anti-tumor Activity of Histone Deacetylase Inhibitor LAQ824” (Manuscript ID: ijms-2546147).. We really appreciate all your helpful comments and suggestions, which will prove invaluable in revising and improving our paper. Based on the instructions, the file of the revised manuscript was uploaded. Accordingly, we have uploaded a copy of the original manuscript with all the changes highlighted by using the track changes mode in MS Word.
We have carefully studied your suggestion point to point and revised the manuscript accordingly. The amendments are listed as follows:
Specific comment 1:
I suggest a revised discussion section that centers on the use of HDACis and how NOTCH inhibition can be achieved, considering the existing drug LBH589 (Panobinostat) and NOTCH inhibitors. This approach will help address readers' concerns and highlight the potential benefits of the combination therapy in a HDACis centric manner.
Response:
Thank you very much for your constructive comments. We have made some changes to the content of the discussion section. The feasibility and prospects for combination of other HDACis and Notchi are emphasised. You can detect it in lines 413-421.
As follows: Activation of the Notch signaling pathway gives a new inspiration to study the toxic effects of HDACis. The relationship between the toxic effects of HDACis and the activation of NOTCH signaling can be further investigated. These study opens up a new direction in the combination therapy of HDACis to cancer. Numerous HDACis also have been found to upregulate the Notch signaling pathway, such as LBH589 (Panobino-stat), valproic acid (VPA) and suberoyl bishydroxamic acid (SBHA), which expressed stronger antitumor effect. So we could further study other combination of HDACis and Notch inhibitor to exploring more effective combined therapies.
Specific comment 2:
Despite the concerns about LAQ and the possibility of its exclusion from clinical trials, the study's broader relevance remains impressive. However, it is crucial for the authors to provide appropriate references at line 44-46.
Response:
Thank you very much for your careful review. We apologize for the error in the citation. We have cited the correct literature1 at line 44-46.
- de Bono, J.S.; Kristeleit, R.; Tolcher, A.; Fong, P.; Pacey, S.; Karavasilis, V.; Mita, M.; Shaw, H.; Workman, P.; Kaye, S.; et al. Phase I Pharmacokinetic and Pharmacodynamic Study of LAQ824, a Hydroxamate Histone Deacetylase Inhibitor with a Heat Shock Protein-90 Inhibitory Profile, in Patients with Advanced Solid Tumors. Clinical Cancer Research 2008, 14, 6663–6673,
We have uploaded the word version of our manuscript with track changes. We have also uploaded a PDF version of manuscript without track changes.
Thank you very much for your time and kind consideration.
Sincerely yours,
Qinglang Mei
Reviewer 2 Report
the version i downloaded is not very clear.
the authors should have performed experiments on all 9 cell lines with different concentrations, to demonstrate the inhibition of proliferation, based on this first experiment, it was decided which line was more suitable, but in the results I readFurthermore, colony-formation assay analysis of HCT116 cell line indicated that LAQ could inhibit tumor cell proliferation of long-term.
in figure 1 /E was the LAQ incubated at the following concentrationc?
0-- 0.01- 0.005- 0.1 ????
of figure 1/H I wanted to see the RT_PCR image
LAQ concentration is 0.50 or 0.20 I see 2 different values in the G and H graph
figures 7 G and F are not clear, I ask you to re-submit these photographs with a marker and with the wording of the samples are not clear, here I was referring to the Western Blot figures of the supplemental data.
Moderate editing of English language required
Author Response
Dear reviewer:
Thanks very much for the comments to us and for allowing us to revise our manuscript entitled “Inhibition of Notch Signaling Enhances Anti-tumor Activity of Histone Deacetylase Inhibitor LAQ824” (Manuscript ID: ijms-2546147).. We really appreciate all your helpful comments and suggestions, which will prove invaluable in revising and improving our paper. Based on the instructions, the file of the revised manuscript was uploaded. Accordingly, we have uploaded a copy of the original manuscript with all the changes highlighted by using the track changes mode in MS Word.
We have carefully studied your suggestion point to point and revised the manuscript accordingly. The amendments are listed as follows:
Specific comment 1:
the authors should have performed experiments on all 9 cell lines with different concentrations, to demonstrate the inhibition of proliferation, based on this first experiment, it was decided which line was more suitable, but in the results I read. Furthermore, colony-formation assay analysis of HCT116 cell line indicated that LAQ could inhibit tumor cell proliferation of long-term.
in figure 1 /E was the LAQ incubated at the following concentrationc?
0-- 0.01- 0.005- 0.1 ????
Response:
I apologize for the lack of clarity in the description of the results. We first tested the IC50 value of LAQ on 9 different cell lines. It was found that HCT116 was the most sensitive to LAQ with the lowest IC50 value. So HCT116 was chosen for the follow-up study. Since the IC50 value of HCT116 was about 0.05 μM, we chose 0.05 μM as the intermediate concentration, and selected 0.01 μM and 0.1 μM as the low and high concentrations for the assay, respectively. We have revised the manuscript for more detailed descriptions, at line 177.
As follows: So we chose the HCT116 cell line (IC50 = 0.05 μM) for subsequent experiments.
Specific comment 2:
of figure 1/H I wanted to see the RT_PCR image
Response:
Here we will upload the raw images of qRT-PCR and give a histogram of the CT difference between P21 and GAPDH.
Specific comment 3:
LAQ concentration is 0.50 or 0.20 I see 2 different values in the G and H graph
Response:
Thank you for your careful review. The aim of Fig 1H,G was to test whether there is a dose-dependent inhibition of histone acetylation by LAQ. According to the IC50(0.05μM) values of LAQ for HCT116 cell line, the experimental concentrations of Fig1G,H were setting. We focused on detecting the associated protein and mRNA expression after treatment of HCT116 cells at concentration of 0μM , 0.01μM, 0.05μM and 0.1μM. However, due to the amount of drug available, the concentration was subsequently adjusted to 0.2μM not 0.5μM. There was no significant effect on the focus of our experiments. To reduce the confusion, we have removed the results with a concentration of 0.2 μM.
Specific comment 4:
figures 7 G and F are not clear, I ask you to re-submit these photographs with a marker and with the wording of the samples are not clear, here I was referring to the Western Blot figures of the supplemental data.
Response:
Thank you for reviewing the article. Here I am attaching the original picture of the WB. The image includes the marker and strip names as well as the sample name. I hope it will meet your requirements. Thanks.
Fig. 7F
Fig.7G
We have uploaded the word version of our manuscript with track changes. We have also uploaded a PDF version of manuscript without track changes.
Thank you very much for your time and kind consideration.
Please see the picture in the attachment.
Sincerely yours,
Qinglang Mei

Reviewer 3 Report
This manuscript is devoted to an interesting topic - enhancing the effectiveness of antitumor drugs, elucidating the multitarget mechanisms of their action.
The authors obtained interesting results, which testify to new approaches in antitumor therapy, opening up the possibility of a more effective impact.
In general, the article is written clearly and understandably, accessible to the reader. I have a few comments that I would like the authors to take into account.
A few general points:
1) According to the logic of the development of the article, I would like to get an answer whether the activation of the discussed signaling pathway is associated with the toxicity that caused the termination of clinical trials. In conclusion, the authors cover this issue in a very veiled way.
2) very often observed effects are characterized qualitatively, not quantitatively. I would like to clarify the term "significantly". By what percentage is the volume of the tumor reduced? Or how much lower dose of LAQ can be used to achieve the same antitumor effect.
3) the rationale for the choice of doses of LAQ and PF, as well as the rationale for the choice of the method of administering them to animals, should be added to the text of the manuscript.
Also a few minor notes:
1) Explanation of many abbreviations is not given at their first mention
2) I know what hydroxamic acid is, but neither I nor Google know what isohydroxamic acid is (line 164).
3) lines 360-362 - repeating the same phrase twice
4) section 6 should be deleted
5) I would like to see the structure of the PF inhibitor
Author Response
Dear reviewer:
Thanks very much for the comments to us and for allowing us to revise our manuscript entitled “Inhibition of Notch Signaling Enhances Anti-tumor Activity of Histone Deacetylase Inhibitor LAQ824” (Manuscript ID: ijms-2546147).. We really appreciate all your helpful comments and suggestions, which will prove invaluable in revising and improving our paper. Based on the instructions, the file of the revised manuscript was uploaded. Accordingly, we have uploaded a copy of the original manuscript with all the changes highlighted by using the track changes mode in MS Word.
We have carefully studied your suggestion point to point and revised the manuscript accordingly. The amendments are listed as follows:
Specific comment 1:
According to the logic of the development of the article, I would like to get an answer whether the activation of the discussed signaling pathway is associated with the toxicity that caused the termination of clinical trials. In conclusion, the authors cover this issue in a very veiled way.
Response:
Thank you very much for your constructive comments. We feel very sorry for the lack of detail in the description. Whether the toxicity of LAQ leading to the termination of clinical trials is related to NOTCH activation still needs further study. Our results suggest that activated NOTCH affects the anti-tumor efficacy of LAQ, and the toxicity of LAQ can be reduced by combination of PF. It might be possible to reduce the dosage of LAQ with the same antitumor effect and reduce the toxicity by combination of PF. We have explained this in the discussion section, at line 400-402.
As follows: Activation of the Notch signaling pathway gives a new inspiration to study the toxic effects of HDACis. The relationship between the toxic effects of HDACis and the activation of Notch signaling can be further investigated.
Specific comment 2:
very often observed effects are characterized qualitatively, not quantitatively. I would like to clarify the term "significantly". By what percentage is the volume of the tumor reduced? Or how much lower dose of LAQ can be used to achieve the same antitumor effect.
Response:
Thank you very much for your suggestion. We have added the appropriate significant values to the manuscript. and the dosage of LAQ be used alone to achieve the same antitumor effect, at line 275-278.
As follows: Results showed that LAQ (10 mg/kg) delayed tumor growth and inhibition rate for tumor development was 43.90%. When combined usage of LAQ (10 mg/kg) and PF (100 mg/kg), inhibitions effect of tumor growth was 63.69% (Fig. 6E), which is equal to the inhibitory effect of LAQ (15 mg/kg).
Specific comment 3:
the rationale for the choice of doses of LAQ and PF, as well as the rationale for the choice of the method of administering them to animals, should be added to the text of the manuscript.
Response:
Thank you for your suggestions, we have added the dosing and references in the manuscript. You can find them in lines 271-274.
As follows: According to the research previously published and the metabolic dynamics of the chemicals, we selected an intravenous dose scheme of 10 mg/kg of LAQ and an oral dosage of 100 mg/kg of PF, with their respective tumor growth inhibition rates of up to 40%.
Specific comment 4:
Explanation of many abbreviations is not given at their first mention
Response:
Thank you very much for your valuable comments. We have scrutinized the abbreviations, attaching explanations to the first occurrence.
As follows: half-maximal inhibitory concentration (IC50); Hes family bHLH transcription factor 1(Hes1); hes related family bHLH transcription factor with YRPW motif 1(Hey1) and soon on. We have marked the changes using highlighting in the manuscript.
Specific comment 5:
I know what hydroxamic acid is, but neither I nor Google know what isohydroxamic acid is (line 164).
Response:
Thanks to your careful reading, we have added explanations of isohydroxamic acid in line 167-168. It has geometric isomers mainly because of the asymmetry of the oxime group. It is mainly dominated by isohydroxamic acid.
As picture follows: The isohydroxamic acid is the reciprocal isomers of hydroxamic acid.
hydroxamic acid(R-C=N-OH)
isohydroxamic acid(R-(C=O)-N-OH)
Specific comment 6:
lines 360-362 - repeating the same phrase twice
Response:
Thank you for your careful reading. We apologize for not checking the manuscript carefully enough. We have removed the phrase of repeating “apoptotic differentiation”, which you can see in line 360-362.
As follows: HDACis not only exerts direct antitumor effects by regulating the acetylation status in tumor cells, inducing apoptosis, blocking cell cycle, promoting DNA damage and inducing autophagy, but also affects other signaling pathways through epigenetic pathways and thus influences the effects on tumor cells.
Specific comment 7:
section 6 should be deleted
Response:
Thank you for your suggestion. We have deleted the sixth section from the manuscript and put it in the supplementary material Fig. S3. You can see it in the new version of our manuscript.
Specific comment 8:
I would like to see the structure of the PF inhibitor
Response:
Thank you for your suggestion. We have added the structure of the PF inhibitor in Figure. S2A in supplement material. And you can see the structure of PF below.
We have uploaded the word version of our manuscript with track changes. We have also uploaded a PDF version of manuscript without track changes.
Thank you very much for your time and kind consideration.
Please see picture in the attachment.
Sincerely yours,
Qinglang Mei

Round 2
Reviewer 2 Report
Dear Qinglang Mei and others
,
to be published you must add what is required
the work is interesting but to date you still have to elaborate some parts such as:
1.point .
the authors write in the answer as follows :
First tested the IC50 value of LAQ on 9 different cell lines.
the authors write that they tested the nine cell lines, write in the results for verify the killing effect on solid tumor cells, cell viability were assessed by applying linear dosages of LAQ to different solid cancer cell lines, including breast cancer, lung cancer, and colorectal malignancies.
I ask the authors the linear doses of LAQ applied to the various cell lines to which concentration do they correspond?
2 point
. the authors did not answer the following question ?
in figure 1 /E was the LAQ incubated at the following concentrationcs?
0-- 0.01- 0.005- 0.1 ????
regarding the figure E
the authors write in the work ____ “HCT116 cell line treated with 50 nM LAQ for one hour at line 182
instead in the figure the cells are treated at various times?
the authors also write it in the legend of figure 1 E
Legend [Figure 1 (E) Western blot analysis of the Ac-H3 in HCT116 cell line with LAQ for different time, total actin were similar analyzed]
Because to the 50 nM LAQ ???
3 point in the sent file original images ( figure 7 G e 7 F)
regarding the figure Western Blots of the supplemental data.
I see some images of the Western Blots are not clear without a reference marker and without the name of the loaded samples
The authors uploaded samples without writing the name of the sample loaded into the well, if to date you don't remember which sample you loaded into the various Western Blots you sent me, it is absolutely correct not to present the images of the various Western Blots
Moderate editing of English language required
Author Response
Dear reviewer:
Thanks very much for the comments to us and for allowing us to revise our manuscript entitled “Inhibition of Notch Signaling Enhances Anti-tumor Activity of Histone Deacetylase Inhibitor LAQ824” (Manuscript ID: ijms-2546147).. We really appreciate all your helpful comments and suggestions, which will prove invaluable in revising and improving our paper. Based on the instructions, the file of the revised manuscript was uploaded. Accordingly, we have uploaded a copy of the original manuscript with all the changes highlighted by using the track changes mode in MS Word.
We have carefully studied your suggestion point to point and revised the manuscript accordingly. The amendments are listed as follows:
Specific comment 1:
the authors write in the answer as follows :
First tested the IC50 value of LAQ on 9 different cell lines.
the authors write that they tested the nine cell lines, write in the results for verify the killing effect on solid tumor cells, cell viability were assessed by applying linear dosages of LAQ to different solid cancer cell lines, including breast cancer, lung cancer, and colorectal malignancies.
I ask the authors the linear doses of LAQ applied to the various cell lines to which concentration do they correspond?
Response1
We apologize that we didn't understand your question accurately last time.
We have put the linear concentration of LAQ (100, 50, 25, 12.5, 6.25, 3.125, 1.5625, 0.7812, 0.3906, 0.1953, 0.0976, 0.0488, 0.0244, 0.0122, 0.0061 03516, 0.0030, 0.0015, 0.0007 μM) into the legend of Fig. 1C.
As follws:
MTT assay detects proliferation ability of nine different cell lines after 48 hours of treatments with different concentration of LAQ (100, 50, 25, 12.5, 6.25, 3.125, 1.5625, 0.7812, 0.3906, 0.1953, 0.0976, 0.0488, 0.0244, 0.0122, 0.0061 03516, 0.0030, 0.0015, 0.0007 μM);
And the IC50 of LAQ in different cell lines is shown below. And We added relevant concentration information in legend of Fig. 1C and Fig S2G
|
Cell line |
IC50(μM) |
|
SW480 |
32.21±0.401 |
|
DLD-1 |
> 100 |
|
HCT-15 |
34.61±0.3514 |
|
RKO |
> 100 |
|
HT-29 |
> 100 |
|
HCT116 |
0.006593±0.8088 |
|
MDA-MB-231 |
29.29±1.414 |
|
MCF7 |
>100 |
|
A549 |
0.01003±1.076 |
In order to determine the IC50 values of the nine cell lines, we examined the viability of the cell lines at different concentrations of LAQ (100, 50, 25, 12.5, 6.25, 3.125, 1.5625, 0.7812, 0.3906, 0.1953, 0.0976, 0.0488, 0.0244, 0.0122, 0.0061 03516, 0.0030, 0.0015, 0.0007 μM) to treat the tumour cell lines for 48 h in terms of cell viability with the help of the MTT method. The results showed that HCT116 cell line were the most sensitive to LAQ among these nine cells (IC50=0.006593±0.8088 μM). For this reason, HCT116 cells were selected for subsequent studies in this study.
Specific comment 2:
. the authors did not answer the following question ?
in figure 1 /E was the LAQ incubated at the following concentrationcs?
0-- 0.01- 0.005- 0.1 ????
regarding the figure E
the authors write in the work ____ “HCT116 cell line treated with 50 nM LAQ for one hour at line 182
instead in the figure the cells are treated at various times?
the authors also write it in the legend of figure 1 E
Legend [Figure 1 (E) Western blot analysis of the Ac-H3 in HCT116 cell line with LAQ for different time, total actin were similar analyzed]
Because to the 50 nM LAQ ???
Respone2
We are sorry for didn't describe results clearly enough.
Fig. 1E shows the results of Western Blot after treating HCT116 with 0.05 μM LAQ for different times. The expressions 0, 0.5, 1, 2, 4, 6, and 8 in the Fig. 1E refer to treating HCT116 cells with LAQ (0.05 μM) for 0 hour, 0.5 hour, 1 hour, 2 hour, 4 hour, 6 hour, and 8 hour. "LAQ (hrs)" in Fig. 1E refers to the time of LAQ (0.05 μM) treatment with HCT116 cell line in hours.
We described in line 180-182, "P21 was activated, and acetylation of histone3 (Ac-H3) began to increase in HCT116 cell line treated with 50 nM LAQ for one hour" which is meaning that HCT116 cell line were treated with 0.05 μM LAQ for different times, and it was found that the protein of Ac-H3 and mRNA of P21 began to upregulating after 1 h of LAQ treatment.
We change the expression in lines 180-182 to read “LAQ (0.05 μM) was applied to the HCT116 cell line for different periods of time. The results showed that activation of P21 and increased acetylation of histone 3 (Ac-H3) occurred immediately after 1 h of treatment (Fig. 1E, F).”.
Meanwhile we add the exact concentration of LAQ in legend of Fig. 1E and 1F. As follows:
(E) Western blot analysis of the Ac-H3 in HCT116 cell line with LAQ(0.05 μM) for different time, total actin were similar analyzed; (F) The expression of P21 were detected by qRT-PCR after treatment with LAQ(0.05 μM) for different time;
Actually, we first decided to treat HCT116 cells with 0.05 μM LAQ for different times based on the measured IC50 values. The study of the relationship between the time effect of Ac-H3 and P21 with LAQ was performed. As a result, we found that the expression of Ac-H3 and P21 was up-regulated by LAQ within one hour of treatment of HCT116 cell line.
Specific comment 3:
in the sent file original images ( figure 7 G e 7 F)
regarding the figure Western Blots of the supplemental data.
I see some images of the Western Blots are not clear without a reference marker and without the name of the loaded samples
Respone3
Thank you for your careful review. The marker we used is Thermo scientific Pageruler Prestained Protein Ladder, its item number is 26617, and we have put the official markers' pictures in the original Western Blot file, as follows.
Pageruler Prestained Protein Ladder(180-10KDa)
Fig7F and 7G are the extracted proteins from the tumor tissues of animal experiments, while the experiments of Western Blot were performed to detect the relevant target proteins. Here, NC, PF, LAQ and PF+LAQ correspond to the groups of different administration treatments, respectively. Since we need to present the results of different groups on a same gelogram, we showed only three mice per group in the gel. The number of 1, 2, and 3 in the raw data correspond to the individual duplicate animals in our different groups, respectively. In order to express the authenticity of our Western Blot results, we spliced the protein bands with marker according to the place of membrane cutting. You can see the integrity of our whole membrane by the size order of the marker.
Here we configured the gel for 15 wells. We use arrows to one-to-one match each well to the sample name. We carefully modified all of the original Western Blot images. You can see in our new supplement data.
When there are fewer samples, we run Western Blot with samples from other people whose data is not currently available in our lab. For this reason, the sample names of the protein gel wells in some of the raw data plots are not listed in detail. We can only describe the purpose of the assay in general, the exact compound name and cell line can’t be given explicitly. We hope you will understand.
The image of western blot with marker is shown below
Fig. 7F (Fig. 6F now)
Fig. 7G (Fig. 6G now)
We have uploaded the word version of our manuscript with track changes. We have also uploaded a PDF version of manuscript without track changes.
You can view the full answer including pictures in the word document below.
Sincerely yours,
Qinglang Mei
